# CDX2- and PAX8-Expressing Subtypes in Female Urethral Adenocarcinoma: Pathogenesis Insights through Immunohistochemical and Morphological Analyses

**DOI:** 10.3390/diagnostics13142408

**Published:** 2023-07-19

**Authors:** Emiko Sugawara, Yasuyuki Shigematsu, Gulanbar Amori, Keisuke Sugita, Junji Yonese, Kengo Takeuchi, Kentaro Inamura

**Affiliations:** 1Division of Pathology, The Cancer Institute, Japanese Foundation for Cancer Research, Tokyo 135-8550, Japan; emiko.sugawara@jfcr.or.jp (E.S.); yasuyuki.shigematsu@jfcr.or.jp (Y.S.); obulhasim.gulanbar@jfcr.or.jp (G.A.); keisuke.sugita@jfcr.or.jp (K.S.); kentakeuchi-tky@umin.net (K.T.); 2Department of Pathology, The Cancer Institute Hospital, Japanese Foundation for Cancer Research, Tokyo 135-8550, Japan; 3Department of Genitourinary Oncology, The Cancer Institute Hospital, Japanese Foundation for Cancer Research, Tokyo 135-8550, Japan; jyonese@jfcr.or.jp; 4Pathology Project for Molecular Targets, The Cancer Institute, Japanese Foundation for Cancer Research, Tokyo 135-8550, Japan

**Keywords:** CDX2, female urethral adenocarcinoma, immunohistochemistry, PAX8, urologic pathology

## Abstract

Female urethral adenocarcinoma has attracted attention as a rare tumor type based on its differential pathogenesis from its male counterpart. However, to date, our knowledge concerning its immunohistochemical and morphological characteristics remains limited due to the small number of cases studied. In this study, nine consecutive cases of female urethral adenocarcinoma were used for immunohistochemical and morphological characterization of the tumor based on semi-comprehensive immunohistochemical analysis and detailed morphological evaluations. Our immunohistochemical assay revealed two subtypes of female urethral adenocarcinoma with distinctive staining patterns: the CDX2- and PAX8-expressing subtypes. The former stained positive for other intestinal markers (e.g., HNF4α and TFF1) as well (7 of 7 cases); the latter stained negative for these intestinal markers (0 of 2 cases) but stained positive for clear cell carcinoma markers (e.g., Napsin A and HNF1β) (2 of 2 cases). Regarding cytokeratins, the former displayed a CK7- and CK20-positive immunoprofile (7 of 7 cases); the latter exhibited a CK7-positive and CK20-negative immunoprofile (2 of 2 cases). Morphologically, CDX2- and PAX8-expressing subtypes resembled intestinal-type adenocarcinoma and clear cell carcinoma (occurring in gynecological organs), respectively. The semi-comprehensive immunoprofiling data presented in this study can potentially contribute to the correct diagnosis of this rare tumor type. Finally, our study represents an important basis for future investigations aiming to further elucidate the details and origin of female urethral adenocarcinoma, and it can potentially contribute to developing diagnostic and therapeutic strategies for treating this malignancy.

## 1. Introduction

Female urethral carcinoma is a rare malignancy, accounting for <1% of all cancer types in women [1,2]. This tumor exhibits a wide range of morphological traits and is divided into three histological subtypes: urothelial carcinoma, squamous cell carcinoma, and adenocarcinoma [3,4,5]. Unlike bladder tumors, urethral tumors differ both clinically and morphologically by sex, reflecting the distinct anatomic and histologic differences between the male and female urethra [3,6]. For example, in urethral cancer, the prevalence of both urothelial carcinoma and squamous cell carcinoma is higher in men than in women, whereas adenocarcinoma is more frequent among women. Adenocarcinoma accounts for approximately 10–30% of female urethral carcinoma cases, although estimates vary across the literature [3,4,5]. The tumor stage of female urethral adenocarcinoma has traditionally been determined using the same classification as its male counterpart [7]; however, a recent study suggested that it should be determined using female-specific criteria that consider the female urethral anatomy [6]. These findings have substantially increased interest in the development of new diagnostic and therapeutic strategies for managing urethral adenocarcinomas in women.

Because of the rarity of female urethral adenocarcinoma, little information is available about its immunohistochemical and morphological features. This malignancy has been the subject of relatively few investigations, most of which have focused on single to several cases [8,9,10,11,12]. Even though a few studies have examined a relatively large number of cases [1,4,6], none of them performed any exhaustive immunohistochemical or rigorous morphological characterization Therefore, we designed a study with the aim to obtain a complete picture of its immunohistochemical and morphological features. To the best of our knowledge, this study is the first to perform semi-comprehensive immunohistochemical and detailed morphological evaluations of consecutive cases of female urethral adenocarcinoma. This study will help correctly diagnose this rare tumor type and represent an important basis for future investigations on the details and origin of female urethral adenocarcinoma.

## 2. Materials and Methods

### 2.1. Tissue Samples and Patient Data

We enrolled nine consecutive Japanese female patients with urethral adenocarcinoma. All patients had undergone tumor resection between January 1995 and November 2020 at the Cancer Institute Hospital, Japanese Foundation for Cancer Research (JFCR), Tokyo, Japan. No patient had received prior therapy before the tumor resection. The study protocol was approved by the ethical committee of the JFCR (approval number 2018-1177). The requirement for informed consent specific to this study was waived because of the retrospective nature of this study.

### 2.2. Morphological Evaluation

Using 4 μm thick hematoxylin-and-eosin stained, formalin-fixed, and paraffin-embedded (FFPE) tissue sections, we determined whether each tumor displayed the following morphological features: clear cytoplasm [13], nuclear pleomorphism, columnar cell appearance, hobnail cell or signet cell appearance [14], papillary or tubular growth, cribriform pattern, trabecular structure, intracellular or extracellular mucin production, background intestinal metaplasia or endometriosis, necrosis, calcification, and psammoma bodies.

### 2.3. Immunohistochemical Analysis

Immunohistochemistry was performed on 4 μm thick FFPE sections. The following primary antibodies were used: PAX8 (BC12, diluted 1:500; Abcam, Cambridge, UK); CK7 (OVTL12/30, diluted 1:200; Dako, Carpinteria, CA, USA); CK20 (IT-Ks20.8, diluted 1:100; PROGEN Biotechnik, Heidelberg, Germany); CDX2 (DAK-CDX2, diluted 1:100; Dako); MUC2 (Ccp58, ready to use; Leica Biosystems, St. Louis, MO, USA); GATA3 (HG3.31, diluted 1:50; Santa Cruz Biotech, Santa Cruz, CA, USA); PSAP (PASE/4LJ, diluted 1:3000; Dako); Uroplakin 2 (BC21, ready to use; Nichirei, Tokyo, Japan); ER (SP1, ready to use; Roche Tissue Diagnostics, Tucson, AZ, USA); PgR (IE2, ready to use; Roche Tissue Diagnostics); p63 (4A4, diluted 1:100; Abcam); HNF4α (H1415, diluted 1:400; Perseus Proteomics, Tokyo, Japan); Napsin A (IP64, diluted 1:800; Leica Biosystems); HNF1β (polyclonal, diluted 1:500; Sigma-Aldrich, St. Louis, MO, USA); CD10 (56C6, diluted 1:100; Leica Biosystems); p16 (E6H4, ready to use; Roche Tissue Diagnostics); MUC5AC (CLH2, ready to use; Leica Biosystems); TFF1 (polyclonal, diluted 1:800; GeneTex, Irvine, CA, USA); Glypican 3 (1G12, ready to use; Nichirei); SATB2 (SATBA4B10, diluted 1:200; Santa Cruz Biotech); WT1 (WT49, diluted 1:30; Leica Biosystems); TTF1 (8G7G3/1, diluted 1:1000; Dako); HER2 (4B5, ready to use; Roche Tissue Diagnostics); EGFR (EGFR.113, diluted 1:20; Leica Biosystems); and PD-L1 (SP142, ready to use; Roche Tissue Diagnostics). Immunohistochemical staining was performed using Roche Benchmark ULTRA (Roche Diagnostics, Mannheim, Germany) automated system for ER, PgR, HER2, p16, and PD-L1, and Leica Bond III automated system (Leica Microsystems, Buffalo Grove, IL, USA) for the others. The staining intensity (absent, 0; weak, 1; moderate to strong, 2) was assessed to calculate the H-score (1 to 200), defined as the product of the percentage of immunopositive tumor cells and the staining intensity. Three different cut-off values were used: −, +/−, and +, assigned to H-scores of 0, 1–10, and >10, respectively.

## 3. Results

### 3.1. Clinical Characteristics of Female Urethral Adenocarcinoma

Table 1 summarizes the clinical features of the nine cases. The patient age ranged from 33 to 72 years. Of the nine patients, seven had tumors in the distal urethra, one in the proximal urethra, and one over the whole urethra. During the 2- to 16-year follow-up period, four patients had post-surgery recurrences, and two died due to the recurrence.

### 3.2. Immunohistochemical Characteristics of Female Urethral Adenocarcinoma

The immunohistochemical features of the nine cases are presented in Table 2. CDX2, a nuclear transcription factor for intestinal differentiation [15], was expressed in seven out of nine cases (77%) (Figure 1a). PAX8, a nuclear transcription factor associated with organogenesis of the thyroid gland, kidney, and Müllerian system [16], was expressed in two out of nine cases (22%) (Figure 2a). All CDX2-expressing tumors were PAX8-negative, and all PAX8-expressing tumors were CDX2-negative, indicating a mutually exclusive relationship between CDX2 and PAX8 expression. The CDX2-expressing tumors were all positive for CK7 (Figure 1b), CK20 (Figure 1c), HNF4α (Figure 1d), and TFF1 (7/7, 100%) and frequently positive for other intestinal markers, including SATB2 (4/7, 57%), MUC2 (6/7, 85%), and MUC5AC (3/7, 42%). The PAX8-expressing tumors were all positive for CK7 (Figure 2b), Napsin A (Figure 2c), and HNF1β (2/2, 100%) and negative for CK20 (Figure 2d), HNF4α, SATB2, TFF1, MUC2, and MUC5AC (0/2, 0%). Appendix A summarizes the immunohistochemical analysis results for CK7, CK20, CDX2, HNF4α, SATB2, MUC2, MUC5AC, TFF1, PAX8, Napsin A, HNF1β, CD10, GATA3, p16, PSAP, Glypican 3, Uroplakin 2, WT1, TTF1, ER, PgR, HER2, EGFR, and PD-L1 expression.

### 3.3. Morphological Characteristics of Female Urethral Adenocarcinoma

Table 3 shows the morphological features of the nine cases. The CDX2-expressing and PAX8-expressing tumors displayed their own unique set of morphological characteristics.

The CDX2-expressing tumors showed a wide range of morphological characteristics of intestinal-type adenocarcinomas with variable degrees of differentiation. Well-differentiated components formed tubular or papillary structures (Figure 1e,f), moderately differentiated components exhibited a cribriform pattern (Figure 1g), and poorly differentiated components exhibited intra- and extracellular mucin production (Figure 1h).

The PAX8-expressing tumors morphologically resembled gynecological organ-derived clear cell carcinoma. In all PAX8-expressing tumors (2/2), hobnail-like tumor cells with clear cytoplasm formed tubular or papillary patterns (Figure 2e,f).

Abundant mucin was present in 71% (5/7) of the CDX2-expressing tumors and absent in the PAX8-expressing tumors (0/2, 0%). Columnar tumor cells were observed in all and one of the CDX2-expressing (7/7, 100%) and PAX8-expressing tumors (1/2, 50%), respectively. Clear cytoplasm and hobnail-like cell features were observed in all PAX8-expressing tumors (2/2, 100%) but none of the CDX2-expressing tumors (0/7, 0%).

Regarding metaplastic change in the background epithelium, intestinal metaplasia was observed in one of the seven CDX2-expressing tumor cases (1/7, 14%) but none of the PAX8-expressing cases (0/2, 0%). Endometriosis was not evident in any of the nine cases (0/9, 0%).

## 4. Discussion

Female urethral adenocarcinoma has gained increased attention as a rare and distinct neoplasm, owing to its unique pathogenesis compared with its male counterpart. Because of the limited number of cases studied thus far, the immunohistochemical and morphological features of this malignancy remain inadequately understood. In the present study, we conducted an in-depth analysis of a consecutive case series of female urethral adenocarcinoma, employing a comprehensive immunohistochemical approach accompanied by meticulous morphological evaluations. Our findings unveiled two discrete subtypes of female urethral adenocarcinoma: those expressing CDX2 and those expressing PAX8. These subtypes exhibit distinguishing immunohistochemical and morphological features, which may be associated with their respective tumor origins. Our findings contribute to a better understanding of female urethral adenocarcinoma pathogenesis by suggesting that this malignancy encompasses a heterogeneous group of diseases, potentially arising through distinct molecular pathways. In addition, the comprehensive immunoprofiling data generated for female urethral adenocarcinoma may facilitate accurate diagnosis and classification of this rare neoplasm.

The pathological features of female urethral adenocarcinoma have not yet been extensively studied on a large scale. However, there exist several case reports that describe its immunohistochemical and morphological features and propose their potential associations with tumor origins. For example, in a CDX2-expressing female urethral adenocarcinoma case [17,18], intestinal metaplasia was observed in the background epithelium adjacent to the tumor, suggesting a potential origin from intestinal metaplasia. Although our study also identified intestinal metaplasia in one of seven CDX2-expressing cases, limited tissue samples hindered an investigation into background epithelium in some cases. Mehra et al. reported a case study of female urethral adenocarcinoma resembling clear cell carcinoma of the female genital tract and displaying positive immunostaining for CK7 and PAX8 [19], consistent with the PAX8-expressing tumors in our study. Further investigation revealed potential gene fusions, including ANKRD28–FNDC3B, in the PAX8-expressing tumors [19]. This type of adenocarcinoma is assumed to be of Müllerian origin, although this hypothesis is still debated and lacks definitive evidence [4,20]. Skene’s gland adenocarcinoma of the female urethra has been reported in case reports [21,22,23,24]. Skene’s gland is homologous to the male prostate and localized in the distal urethra, and Skene’s gland adenocarcinoma is characterized by positive immunostaining for PSA and NKX3.1, similar to male prostate adenocarcinoma. None of our nine cases showed positive immunostaining for PSA or NKX3.1. Mesonephric (Wolffian-derived) adenocarcinoma of the female urethra, characterized by positive PAX8, GATA3, and CD10 (luminal pattern) expressions, has been recently reported [25]. Neither of our PAX8-expressing tumors showed GATA3 positivity or CD10 luminal staining, although we detected CD10 cytoplasmic staining in one case.

The potential origin of female urethral adenocarcinoma includes intestinal metaplasia of the urothelium, Müllerian duct, Skene’s gland, Wolffian duct, and nephrogenic adenoma [22,25,26,27,28,29,30,31,32]. Because urethral cancer often occurs in the urethral diverticula, which can facilitate intestinal metaplasia of the urothelium, chronic inflammation in the diverticular may promote the development of female urethral adenocarcinoma [33,34,35,36]. The immunostaining results, along with morphological features, revealed two subtypes of female urethral adenocarcinomas, namely CDX2- and PAX8-expressing subtypes. CDX2-expressing tumors appear to arise from the intestinal metaplasia of the urothelium, whereas PAX8-expressing tumors potentially originate from the Müllerian duct. These results support the utility of immunohistochemistry, and the close relationship between the morphological characteristics and immunohistochemical profiles evokes a potential for morphological evaluations in hypothesizing the origin of female urethral adenocarcinoma. Intestinal-type adenocarcinomas arise from various organs and exhibit both intestinal marker expression (e.g., CDX2 and SATB2) and intestinal morphologies. For example, pulmonary enteric-type adenocarcinoma exhibits positive immunostaining for CDX2 and shares morphological characteristics with colorectal adenocarcinoma [37]. Further research with a large sample size is warranted to fully elucidate the distinct subtypes of female urethral adenocarcinoma and their potential origins.

Our study builds on previous research that has explored the role of CDX2 and PAX8 expressions in female urethral adenocarcinoma, as summarized in Table 4 [17,18,19,23,24,38]. CDX2-expressing tumors were characterized by mucinous features or intestinal-type morphology. On the other hand, one PAX8-expressing tumor resembled clear cell adenocarcinoma of the female genital tract similar to our PAX8-expressing tumors [19], whereas another PAX8-expressing tumor was positive for CDX2 and showed intraluminal mucin secretion [38]. The implications of CDX2 and PAX8 expression are still under investigation, and the roles of these transcription factors in female urethral adenocarcinoma need to be further investigated.

Immunostainings are often helpful in identifying the primary site of an unknown primary cancer [39,40,41,42]. The combination of CK7 and CK20 immunostainings aids in localizing the primary site. For example, urothelial carcinoma, especially the luminal type, commonly displays CK7-positive and CK20-positive profiles. In addition, nuclear transcription factors can assist in determining the primary site. For example, CDX2 positivity indicates intestinal differentiation or origin, while PAX8 positivity supports origin in the thyroid gland, kidney, or Müllerian system. Therefore, a CK7-positive/CK20-positive/CDX2-positive or CK7-positive/CK20-negative/PAX8-positive immunoprofile of female urethral adenocarcinoma can serve as a diagnostic clue while searching for the primary site of an unknown primary cancer.

To advance this study, further investigations and addressing potential limitations are necessary. Additional cases of female urethral adenocarcinoma, such as Skene’s gland- or Wolffian duct-derived and diverticulitis-associated tumors, should be examined to compare their immunohistochemical and morphological features with CDX2-expressing and PAX8-expressing subtypes. The molecular characteristics should be investigated for each subtype to provide molecular-based evidence of tumor origins or distinctions. The prognostic relevance and response to specific treatments should also be examined for each subtype, as subtype-specific strategies may better treat the disease if prognosis or response to particular therapies differs. However, this study had some limitations, including a relatively small patient population, which precludes statistical assessment of the results, despite being the largest study to perform semi-comprehensive immunoprofiling and detailed morphological evaluations of this tumor. Additionally, the restricted investigation of the background epithelium might hinder the identification of intestinal metaplasia or Müllerian benign lesions, such as endometriosis. Thorough investigation of background tissues is necessary in future research.

Inflammation is increasingly recognized as a key player in various stages of tumor progression, including initiation, malignant transformation, and metastasis [43,44]. In the context of urethral adenocarcinoma, particularly in women, tissue-associated inflammation contributes to these dynamics [4]. Its role in the outcomes of female patients with urethral adenocarcinoma is outside the scope of this work but can potentially provide important insights into the clinical behavior of female urethral adenocarcinoma. We recognize this as a potential limitation, and future research should explore this aspect.

In conclusion, by examining a sequential series of nine female urethral adenocarcinoma cases, we revealed the presence of CDX2- and PAX8-expressing subtypes. CDX2-expressing tumors exhibited expressions of intestinal markers and morphologically resembled intestinal-type adenocarcinoma, implying their development via intestinal metaplasia. PAX8-expressing tumors resembled clear cell carcinoma of the gynecological organs, both immunohistochemically and morphologically, suggesting their derivation from the Müllerian duct. Since several subtypes of female urethral adenocarcinoma are supposed to exist, thorough immunohistochemical and morphological studies on a high number of cases are warranted to validate our findings and provide insights into tumor origins.

## Figures and Tables

**Figure 1 diagnostics-13-02408-f001:**
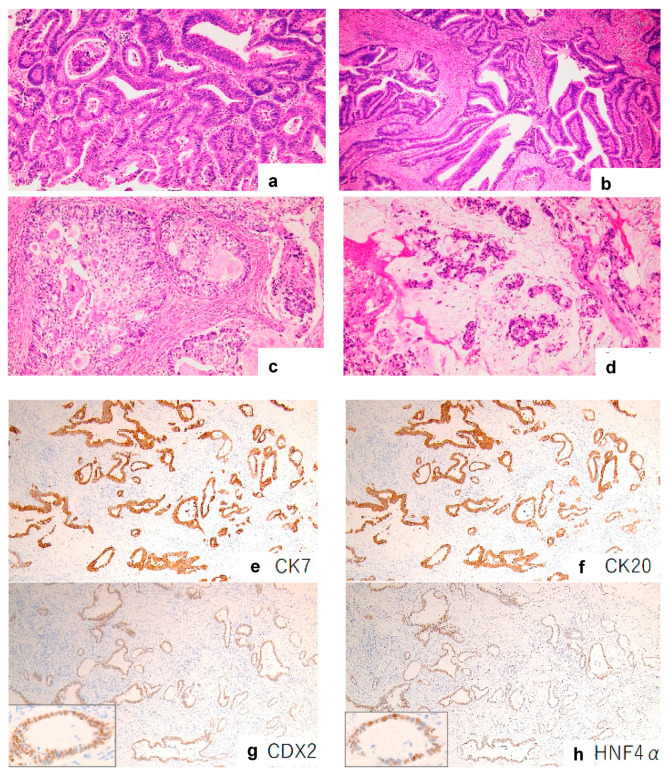
Morphological and immunohistochemical images of CDX2-expressing female urethral adenocarcinoma. Well-differentiated components forming tubular or papillary structures (**a**,**b**), moderately differentiated components exhibiting cribriform patterns (**c**), and poorly differentiated components producing abundant intracellular and extracellular mucins (**d**). All tumors expressing CDX2 (**g**) were positive for CK7 (**e**), CK20 (**f**), and HNF4α (**h**). (**a**–**d**) Hematoxylin and eosin staining, ×100. (**e**–**h**) Immunohistochemical staining, ×100.

**Figure 2 diagnostics-13-02408-f002:**
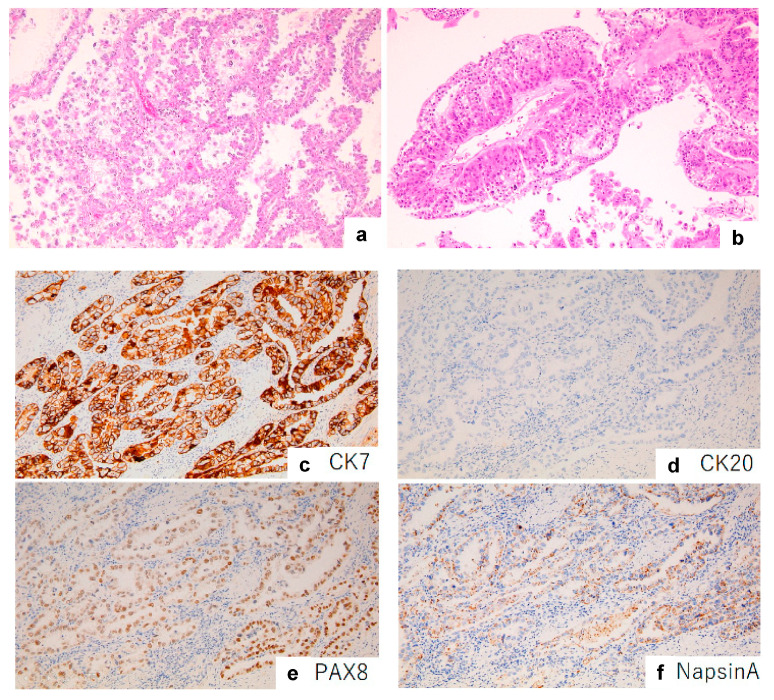
Morphological and immunohistochemical images of PAX8-expressing female urethral adenocarcinoma. Morphologically, hobnail-like tumor cells with clear cytoplasm formed a tubular or papillary pattern (**a**,**b**). All tumors expressing PAX8 (**e**) were positive for CK7 (**c**) and Napsin A (**f**) but negative for CK20 (**d**). (**a**,**b**) Hematoxylin and eosin staining, ×100. (**c**–**f**) Immunohistochemical staining, ×100.

**Table 1 diagnostics-13-02408-t001:** Clinicopathologic features of female urethral adenocarcinoma.

	Age (years)	Follow-Up (years)	Type of Resection	Tumor Location	Tumor Size (mm)	Histology	LVI	LN Metastasis	Pathological Tumor Stage	Outcome
Case 1	62	16	Urethrectomy & cystectomy	Distal, posterior	5	sig > muc > pap	Absent	Absent	pT1	Alive without disease
Case 2	33	5	Local resection	Distal, anterior	16	tub2	Absent	NA	pT2	Distant metastasis to lung (2 years later), died of disease (5 years later)
Case 3	56	13	Anterior pelvic exenteration	Entire	40	tub1 > tub2	Present	Absent	pT4	Distant metastasis to lung (1 year later), alive without disease
Case 4	70	11	Urethrectomy & cystectomy	Distal, anterior	47	pap > muc > sig	Present	Absent	pT3	Alive without disease
Case 5	69	7	Urethrectomy & cystectomy	Distal, anterior	30	muc > sig	Absent	Absent	pT1	Alive without disease
Case 6	72	7	Urethrectomy & cystectomy	Proximal, anterior	38	Clear	Absent	Absent	pT3	Alive without disease
Case 7	63	5	Urethrectomy & cystectomy	Distal, circ	35	pap > tub1 > tub2	Absent	Absent	pT2	Alive without disease
Case 8	47	5	Local resection	Distal, posterior	10	tub1 > tub2	Absent	NA	pTa/is	Local recurrence (5 years later), alive with disease
Case 9	67	2	Urethrectomy & cystectomy	Distal, anterior	15	tub2 with clear cell change	Absent	Present	pT3	Distant metastasis to adrenal gland and brain (2 years later), died of disease (2 years later)

LN, lymph node; LVI, lymphovascular invasion; NA, not available.

**Table 2 diagnostics-13-02408-t002:** Immunohistochemical features of female urethral adenocarcinoma.

	CK7	CK20	CDX2	HNF4a	SATB2	MUC2	MUC5AC	TFF1	PAX8	Napsin A	HNF1β
Case 1	±	±	+	+	±	+	±	+	−	−	±
Case 2	±	+	+	+	±	±	−	±	−	−	−
Case 3	+	+	+	+	+	±	−	±	−	−	±
Case 4	±	+	±	±	−	+	±	±	−	−	−
Case 5	±	+	±	+	−	+	−	±	−	−	−
Case 6	+	−	−	−	−	−	−	−	+	+	+
Case 7	±	+	+	+	−	−	+	+	−	−	−
Case 8	+	±	+	+	+	+	−	±	−	−	−
Case 9	+	−	−	−	−	−	−	−	+	+	±
Total	9/9 (100%)	7/9 (77%)	7/9 (77%)	7/9 (77%)	4/9 (44%)	6/9 (66%)	3/9 (33%)	7/9 (77%)	2/9 (22%)	2/9 (22%)	4/9 (44%)
CDX2- expressing subtype	7/7 (100%)	7/7 (100%)	7/7 (100%)	7/7 (100%)	4/7 (57%)	6/7 (85%)	3/7 (42%)	7/7 (100%)	0/7 (0%)	0/7 (0%)	2/7 (28%)
PAX8- expressing subtype	2/2 (100%)	0/2 (0%)	0/2 (0%)	0/2 (0%)	0/2 (0%)	0/2 (0%)	0/2 (0%)	0/2 (0%)	2/2 (100%)	2/2 (100%)	2/2 (100%)

**Table 3 diagnostics-13-02408-t003:** Morphological features of female urethral adenocarcinoma.

	Case 1	Case 2	Case 3	Case 4	Case 5	Case 6	Case 7	Case 8	Case 9	Total	CDX2- Expressing Subtype	PAX8- Expressing Subtype
Clear cytoplasm	−	−	−	−	−	+	−	−	+	2/9 (22%)	0/7 (0%)	2/2 (100%)
Nuclear pleomorphism	−	−	−	−	−	+	−	−	±	2/9 (22%)	0/7 (0%)	2/2 (100%)
Columnar cell appearance	+	+	+	+	+	−	+	+	±	8/9 (88%)	7/7 (100%)	1/2 (50%)
Hobnail cell appearance	−	−	−	−	−	+	−	−	+	2/9 (22%)	0/7 (0%)	2/2 (100%)
Signet cell carcinoma	++	−	−	+	++	−	−	−	−	3/9 (33%)	3/7 (42%)	0/2 (0%)
Papillary growth	−	++	−	+	−	++	++	−	++	5/9 (55%)	3/7 (42%)	2/2 (100%)
Tubular growth	+	++	+	+	−	+	+	+	+	9/9 (100%)	7/7 (100%)	2/2 (100%)
Cribriform pattern	+	−	+	+	+	−	±	−	−	5/9 (55%)	5/7 (71%)	0/2 (0%)
Trabecular structure	−	−	±	−	±	−	−	−	−	2/9 (22%)	2/7 (28%)	0/2 (0%)
Intracellular mucin production	++	±	+	++	++	−	±	−	−	6/9 (66%)	6/7 (85%)	0/2 (0%)
Extracellular mucin production	++	−	++	++	++	−	+	−	−	5/9 (55%)	5/7 (71%)	0/2 (0%)
Background intestinal metaplasia	−	−	−	−	+	−	−	−	−	1/9 (11%)	1/7 (14%)	0/2 (0%)
Background endometriosis	−	−	−	−	−	−	−	−	−	0/9 (0%)	0/7 (0%)	0/2 (0%)
Necrosis	−	−	+	−	−	+	−	−	−	2/9 (22%)	1/7 (14%)	1/2 (50%)
Calcification	−	−	+	−	−	−	−	−	−	1/9 (11%)	1/7 (14%)	0/2 (0%)
Psammoma body	−	−	−	−	−	−	−	−	−	0/9 (0%)	0/7 (0%)	0/2 (0%)

**Table 4 diagnostics-13-02408-t004:** Previous studies on CDX2 and PAX8 expressions in female urethral adenocarcinoma.

Author(s)	Year	Number of Cases	CDX2 Positive	PAX8 Positive	Morphology
Mehra et al. [19]	2014	1	N/A	Yes (1/1)	Clear cell adenocarcinoma
Satyanarayan et al. [17]	2015	1	Yes (1/1)	N/A	Adenocarcinoma with mucinous features
Harari et al. [18]	2016	5	Yes (4/5)	No (0/5)	All cases were mucinous adenocarcinoma, four of which were CDX2 positive and all were PAX8 negative.
Muto et al. [23]	2017	1	Yes (1/1)	No (0/1)	Adenocarcinoma with mucinous features
Tregnago et al. [24]	2018	1	Yes (1/1)	No (0/1)	Adenocarcinoma with intraluminal mucin secretion
Torrez et al. [38]	2023	1	Yes (1/1)	Yes (1/1)	Adenocarcinoma with intraluminal mucin secretion
Present case	2023	9	Yes (7/9)	Yes (2/9)	CDX2-positive tumors resembled intestinal-type adenocarcinoma. PAX8-positive tumors resembled clear cell carcinoma occurring in gynecological organs.

N/A, not available.

## Data Availability

The datasets used and/or analyzed during the current study are available from the corresponding author on reasonable request.

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
