# Peer review of "CDX2- and PAX8-Expressing Subtypes in Female Urethral Adenocarcinoma: Pathogenesis Insights through Immunohistochemical and Morphological Analyses"

_diagnostics, 2023, doi:10.3390/diagnostics13142408_

Round 1

Reviewer 1 Report

This study aimed to evaluate the immunohistochemical and morphological characterization of female uretheral adenocarcinoma based on semi-comprehensive immunohistochemical analysis and detailed morphological evaluations. I suggest a few addition and modification for the better understanding.

-       Please show the table of the significant studies on CDX2 and PAX8 in urethral adenocarcinoma.

-       Please add the sufficient discussion about it.

 Minor editing of English language required

Reviewer 2 Report

The study is very interesting and gives us important news. The methodology is correct and the conclusions are in line with the study aims. So, I have only one minor comment: please discuss the role of tissue-associated inflammation in the patients' outcomes.

Round 2

Reviewer 1 Report

It's corrected well enough that it can be published.